# Cerebellar Blood Flow and Gene Expression in Crossed Cerebellar Diaschisis after Transient Middle Cerebral Artery Occlusion in Rats

**DOI:** 10.3390/ijms21114137

**Published:** 2020-06-10

**Authors:** Naoya Kidani, Tomohito Hishikawa, Masafumi Hiramatsu, Shingo Nishihiro, Kyohei Kin, Yu Takahashi, Satoshi Murai, Kenji Sugiu, Takao Yasuhara, Ikuko Miyazaki, Masato Asanuma, Isao Date

**Affiliations:** 1Department of Neurological Surgery, Okayama University Graduate School of Medicine, Dentistry and Pharmaceutical Sciences, Okayama 700-8558, Japan; naoya.kidani@gmail.com (N.K.); mhiramatsu@okayama-u.ac.jp (M.H.); n.shingo777@gmail.com (S.N.); thekinkorea@gmail.com (K.K.); takahashiyuu0808@gmail.com (Y.T.); juggler3104@yahoo.co.jp (S.M.); ksugiu@md.okayama-u.ac.jp (K.S.); tyasu37@cc.okayama-u.ac.jp (T.Y.); idate333@md.okayama-u.ac.jp (I.D.); 2Department of Medical Neurobiology, Okayama University Graduate School of Medicine, Dentistry and Pharmaceutical Sciences, Okayama 700-8558, Japan; miyazaki@cc.okayama-u.ac.jp (I.M.); asachan@cc.okayama-u.ac.jp (M.A.)

**Keywords:** apoptosis, cerebral blood flow, crossed cerebellar diaschisis, ischemic stroke, oxidative stress

## Abstract

Crossed cerebellar diaschisis (CCD) is a state of hypoperfusion and hypometabolism in the contralesional cerebellar hemisphere caused by a supratentorial lesion, but its pathophysiology is not fully understood. We evaluated chronological changes in cerebellar blood flow (CbBF) and gene expressions in the cerebellum using a rat model of transient middle cerebral artery occlusion (MCAO). CbBF was analyzed at two and seven days after MCAO using single photon emission computed tomography (SPECT). DNA microarray analysis and western blotting of the cerebellar cortex were performed and apoptotic cells in the cerebellar cortex were stained. CbBF in the contralesional hemisphere was significantly decreased and this lateral imbalance recovered over one week. Gene set enrichment analysis revealed that a gene set for “oxidative phosphorylation” was significantly upregulated while fourteen other gene sets including “apoptosis”, “hypoxia” and “reactive oxygen species” showed a tendency toward upregulation in the contralesional cerebellum. MCAO upregulated the expressions of nuclear factor erythroid 2-related factor 2 (Nrf2) and heme oxygenase-1 (HO-1) in the contralesional cerebellar cortex. The number of apoptotic cells increased in the molecular layer of the contralesional cerebellum. Focal cerebral ischemia in our rat MCAO model caused CCD along with enhanced expression of genes related to oxidative stress and apoptosis.

## 1. Introduction

The term “diaschisis” was coined by von Monakow as the temporary functional shock of intact regions distant to the lesion [1]. Since then, several patterns of diaschisis have been identified, and Baron et al. first described the occurrence of crossed cerebellar diaschisis (CCD) after hemispheric stroke in 1981 [2]. CCD is characterized by decreased metabolism and blood flow in the cerebellar hemisphere contralateral to a supratentorial brain lesion, such as an epileptic lesion, brain trauma, Alzheimer’s disease, or stroke [3]. To date, the mechanism of CCD has been assumed to begin with an interruption of the excitatory input to the cerebellum, mainly via the cortico-ponto-cerebellar pathway, which causes hypometabolism and hypoperfusion of the cerebellum [4,5].

In the previous researches evaluating CCD, most of them were concerned with cerebral ischemic stroke. Some studies indicated that a volume of infarct, a severity of impaired perfusion, a location of stroke, and a side of lesion are associated with CCD [6,7,8,9]. And some scholars showed the association between CCD and functional outcomes [8,10], and showed the significance of CCD as a diagnostic aid of supratentorial impairment and a prognostic factor after revascularization surgery [11,12]. They indicated that CCD is not only an associated phenomenon of supratentorial tissue damage, but an important indicator of the therapeutic effect, functional recovery and prognosis of patients. Furthermore, the recent data based on rodent models of cerebral ischemia suggest that stimulation of the cerebellar nucleus is efficient for post-stroke neurorehabilitation, termed as “upside-down CCD” [13,14].

As CCD has been mainly reported and discussed in clinical settings, little is known about the molecular and biological mechanisms of CCD in either human or experimental models. The preclinical characterization of CCD in animal models might provide a novel understanding of neural interaction after brain injury. In the present study, we evaluated the chronological changes in CCD using single photon emission computed tomography (SPECT) in a rat middle cerebral artery occlusion model, examined gene expression patterns in the cerebellar hemispheres in CCD, and performed histopathological analysis.

## 2. Results

### 2.1. Baseline Characteristics

2,3,5-triphenyltetrazolium hydrochloride (TTC) staining at the endpoint of each protocol revealed cerebral infarction in the middle cerebral artery (MCA) perfusion area, consisting of both cerebral cortex and deep white matter, in all rats subjected to middle cerebral artery occlusion (MCAO). None of the rats in the control group showed any obvious TTC-unstained areas.

### 2.2. Time Course of CBF and CbBF after MCAO

A representative SPECT image coregistered with its corresponding computed tomography (CT) image is shown in Figure 1. Cerebral blood flow (CBF) and cerebellar blood flow (CbBF) perfusion rate were evaluated using N-isopropyl-^123^I-p-iodoamphetamine (^123^I-IMP) SPECT/CT for both control group and MCAO group (Figure 2). Mean CBF r/l ratio of control group was 100.7 ± 2.5%. Compared to the control group, the CBF ratio was significantly decreased two and seven days after MCAO (two days: 36.6 ± 16.9%, seven days: 62.1 ± 20.0%). When the CBF ratio was compared between two and seven days after MCAO, it significantly recuperated at seven days after MCAO. Mean CbBF l/r ratio of control group was 100.1 ± 2.9%. Compared to the control group, the CbBF ratio was significantly decreased two and seven days after MCAO (two days: 85.7 ± 6.6%, seven days: 93.3 ± 2.4%). When the CbBF ratio was compared between two and seven days after MCAO, it significantly rose up at seven days after MCAO.

### 2.3. Gene Expression Changes in the Cerebellar Cortex Induced by MCAO

In the contralateral cerebellar cortex, 765 genes were increased and 1813 were decreased after MCAO as compared to the control group. In the ipsilateral cerebellar cortex, the expression levels of 1183 genes were increased after MCAO while those of 813 genes were decreased. Among the increased genes, 502 were up-regulated in both the contralateral and ipsilateral cerebellar cortex; among the decreased genes, 500 were down-regulated on both sides (Figure 3).

We compared the cerebellar cortices from the MCAO group with those from the control group using gene set enrichment analysis (GSEA) (Table 1 and Table 2). In the present study, GSEA identified 14 gene sets that tended to be upregulated in the cerebellar cortices of the MCAO group, including “hypoxia”, “apoptosis”, and “reactive oxygen species pathway”. At a nominal *p*-value of 5%, only the gene set known as “oxidative phosphorylation” was detected as being significantly enriched. And 30 gene sets were identified as downregulated in the cerebellar cortices of the MCAO group. No gene sets were significantly enriched in the left cerebellar cortices of the MCAO group, with a false discovery rate (*q*-value) of 25%.

### 2.4. Expression of Oxidative Stress-Related Proteins in the Cerebellar Cortex

Given these GSEA results, we next evaluated the expression patterns of the oxidative stress-related factors nuclear factor erythroid 2-related factor 2 (Nrf2) and heme oxygenase-1 (HO-1) in the cerebellar cortex (Figure 4). Western blotting showed that the expression of Nrf2 was upregulated in the cerebellar cortex on the side contralateral to the infarct (0.29 ± 0.04) compared to the cerebellar cortex on the ipsilateral side (0.21 ± 0.04) as well as to the control group (0.20 ± 0.04). The expression of HO-1 was also upregulated on the contralateral side (0.41 ± 0.06) compared to the ipsilateral side (0.28 ± 0.04) and to the control group (0.30 ± 0.03).

### 2.5. Apoptosis in the Cerebellar Cortex Induced by MCAO

To confirm the effect of MCAO on apoptosis in the cerebellar cortex, terminal deoxynucleotidyl transferase (TdT) deoxyuridine triphosphate nick-end labeling (TUNEL) assay was performed. TUNEL+/propidium iodide (PI) + cells were identified only in the molecular layer of the cerebellar cortex (Figure 5). The number of TUNEL+/PI+ cells in the molecular layer was significantly larger in the left hemisphere (contralateral to the cerebral infarct, 11.0 ± 4.7) than in the right hemisphere (ipsilateral to the cerebral infarct, 1.5 ± 0.9).

## 3. Discussion

### 3.1. Cerebral and Cerebellar Blood Flow Analysis

Takuwa et al. have previously evaluated CCD caused by MCAO in mice using laser doppler flowmetry (LDF) [15]. Although LDF can measure CbBF stably and reproducibly, it requires craniotomy of the posterior cranial fossa and can only evaluate the brain surface. Small animal positron emission tomography (PET) provides a higher photon detection sensitivity than SPECT does, but its spatial resolution is usually inferior to that of SPECT [16]. Furthermore, radioactive tracers for SPECT are easier to prepare and handle than those for PET. For these reasons, SPECT is the best modality for detecting CCD in rodent models. In our study, consistent with previous studies [17,18], the CBF in the infarct area notably decreased after the insult and was restored over the following one week. This recovery probably resulted from post-ischemic rebound due to luxury perfusion [19]. The CbBF ratio, i.e., the ratio of CbBF on the contralateral side to that on the ipsilateral side, also decreased in the acute phase of cerebral infarct but did not reach the same degree of lateral asymmetry seen in CBF. This change can be identified as CCD. Furthermore, the CbBF asymmetry also improved in association with the improvement in the CBF asymmetry. This result suggests that the degree of CCD may correlate with supratentorial lesional blood flow. It has been reported in several human studies that diaschisis is potentially reversible if supratentorial reperfusion can be achieved [9,20]. Sobesky et al. have shown that the degree of supratentorial hypoperfusion is correlated with the degree of CCD before and after cerebral reperfusion. Our findings regarding the CbBF time course are in keeping with these previous clinical studies.

### 3.2. Gene Expression in the Cerebellar Cortex

To the best of our knowledge, no previous studies have reported on gene expression in the contralesional cerebellar cortex. Hypoperfusion of the cerebellar hemisphere, which is caused by CCD, might be one of the most important factors affecting the expression of genes related to oxidative stress. In this study, several gene sets associated with inflammation and oxidative stress were identified as upregulated in the contralesional cerebellar cortex. The redox-sensitive transcriptional factor Nrf2 is widely expressed in the central nervous system and is one of the most major regulators of cellular defense mechanisms against oxidative stresses through its coordination of stress-inducible activation of multiple cytoprotective genes [21,22,23,24]. Nrf2 binds to the antioxidant response element sequence and thereby upregulates the expression of its target genes. One of these Nrf2 target genes, HO-1, regulates antioxidant defense and oxidant signaling [25]. Taking these findings together, Nrf2 and HO-1 can be considered as major oxidative stress sensors. We can conclude that Nrf2-regulating anti-oxidative molecules are compensatory induced by CCD to ameliorate oxidative stress.

The gene set of “apoptosis” also tends to be upregulated in the contralateral cerebellar cortex. The expression of these “apoptosis” genes might be triggered by the hypoperfusion induced by CCD. Jie et al. detected caspase-3-positive cells and TUNEL-positive cells in the contralateral cerebellar cortex in the acute phase of MCAO [26]. Similarly, we observed a lateral asymmetry of TUNEL-positive cells, with more TUNEL-positive cells in the contralesional cerebellar cortex. Our results indicate that apoptosis in this case is not caused by a systemic phenomenon, which would affect the cerebral cortex symmetrically. Rather, the asymmetrical apoptosis must be caused by an asymmetrical factor, such as cerebral blood flow and/or input from the cerebrum. The numbers of apoptotic cells counted in our study were much lower than those in previous reports. The reasons for the differences in frequency and location of apoptotic cells between our study and previous reports are unclear, but they may be related to differences in experimental conditions, such as experimental models, timing of the evaluations, and reagents. In the chronic phase of supratentorial injury, TUNEL-positive cells were detected only in the lateral cerebellar nucleus and not in the cerebellar cortex [27]. The cortico-ponto-cerebellar pathway originates from the cerebral cortex and mostly terminates in the cerebellar gray matter [28,29]. Taking these findings together, we can conclude that CCD primarily affects the contralateral cerebellar cortex and that, in the chronic phase, secondary changes may occur in the deep white matter, including the cerebellar nucleus. Since this study could be a preliminary data, further investigation is needed to clarify the molecular mechanisms of deafferentation and their relationship to oxidative stress and apoptosis.

### 3.3. Study Limitations

This study has several limitations. First, our observations were taken only in the very acute phase of MCAO. In clinical practice, CCD sometimes persists even after the reperfusion of the supratentorial ischemic lesion. In future studies, CbBF and protein expression should be observed over a longer period. Second, we have not proved a decrease in the excitatory input to the cerebellum. It would be difficult to be prove this, but a novel method for the visualization of the neural network, such as diffusion tensor imaging, may help. Finally, it remains unclear whether CCD is actually a direct cause of oxidative stress and apoptosis. Deeper insight into the relationship between these phenomena as observed in this study requires further studies with larger sample sizes.

## 4. Materials and Methods

### 4.1. Ethics Statement

This study was conducted in accordance with the guidelines of the Institutional Animal Care and Use Committee of Okayama University Graduate School of Medicine and reported in compliance with ARRIVE (Animal Research: Reporting of In Vivo Experiments) guidelines. The protocol was approved by the Institutional Animal Care and Use Committee of Okayama University Graduate School of Medicine (protocol #OKU-2018179, approved on 5 April 2018). All efforts were made to minimize animal suffering. Measurements and analyses were performed by examiners blinded to the study.

### 4.2. Animals

Adult male Wistar rats (*n* = 57, nine weeks old; SHIMIZU Laboratory Supplies Co., Ltd., Kyoto, Japan) weighing 280 to 300 g were used in this study. They were housed in a temperature-and humidity-controlled room and maintained on a 12-h light/dark cycle with free access to food and water.

### 4.3. Surgical Procedures

Rats were randomly assigned to the experimental or the control group. Transient middle cerebral artery occlusion (MCAO) was carried out according to the intraluminal suture method as previously reported [30,31]. Under general anesthesia (2.0% sevoflurane in 70% N_2_O and 30% O_2_), the bifurcation of the right common carotid artery was exposed. After the right external carotid artery (ECA) was cut, a 4–0 monofilament nylon suture with a silicone-coated tip (Xantopren L blue & ACTIVATOR 2 Universal Liquid, Heraeus Kulzer GmbH & Co. KG, Hanau, Germany) was inserted from the arteriotomy of the ECA toward the origin of the right MCA. After 90 min of MCAO, the filament was withdrawn and the ECA was cauterized. In the control group, the right common carotid artery was also exposed, but arteriotomy was not performed. At the end of the operation, the skin was closed using 3–0 silk sutures. All rats were euthanized with an overdose of pentobarbital (150 mg/kg) at the end of their respective protocols described below. After euthanasia, to confirm cerebral infarction, a coronal brain slice was made 2.0 mm posterior to the bregma and stained using 2,3,5-triphenyltetrazolium hydrochloride (TTC).

### 4.4. Neurological Assessment

The modified Neurological Severity Score (mNSS) was evaluated one day after MCAO. This score was used to assess motor function, sensory disturbance, reflex, and balance. Neurological function was graded on a scale of 0 to 18 (normal score: 0; maximal deficit score: 18) [32]. To uniform a postsurgical neurological severity, only rats that scored between 7 and 12 points on the mNSS one day after MCAO were used in the subsequent experiments. At the time of the neurological assessment, 13 rats were excluded from the subsequent analysis: 3 rats died within one day of MCAO, and 10 showed mNSS under 7.

Eventually, 44 rats were analyzed. Within 44 rats, 29 rats underwent MCAO (8 for blood flow assessment, 4 for cDNA microarray, 6 for western immunoblotting, 11 for TUNEL analysis), and 15 rats were in the control group (5 for blood flow assessment, 4 for cDNA microarray, 6 for western immunoblotting).

### 4.5. Blood Flow Assessment Using SPECT

Two and seven days after MCAO, the eligible rats in the MCAO group (*n* = 8) were scanned using SPECT for small animals with an N-isopropyl-^123^I-p-iodoamphetamine (^123^I-IMP; Iofetamine Injection Daiichi, FUJIFILM RI Pharma Co., Ltd., Tokyo, Japan) tracer to evaluate CBF and CbBF. The rats in the control group (*n* = 5) were likewise scanned two days after their sham surgery. The rats were placed under general anesthesia with 2.0% sevoflurane in a mixture of room air (flow, 2.0 L/min). The scan was performed under general anesthesia in the prone position 15 min after ^123^I-IMP tracer injection (30 MBq) into the lateral tail vein.

Images were obtained using a SPECT/CT scanner (FX3000, TriFoil Imaging Inc., Northridge, CA, USA) with cadmium–zinc–telluride semiconductors and multi-pinhole collimators (focal length, 65 mm; aperture, 1.0 mm). The CT component of the resulting images was used only for anatomical reference in examining the SPECT images. SPECT images were reconstructed using FLEX-RECON software with a three-dimensional ordered subset expectation maximization (iteration, 5; subset, 8) algorithm and data was collected (360° acquisition, 30 s/frame, 64 frames total) with 45-mm semidiameter detectors.

CBF/CbBF imaging was automatically obtained using free analysis software (AMIDE, a Medical Imaging Data Examiner, version 1.0.5) and analyzed semiquantitatively. We performed CBF/CbBF analysis using the region of interest (ROI) settings and calculated the radiation intensity of each ROI automatically (Figure 1). Prolate spheroid shaped ROIs (6 × 9 × 6 mm) were symmetrically placed in the coronal slices 2.0 mm posterior to the bregma and were used in analyzing CBF in the MCA territory. The cerebral perfusion ratio was obtained according to the following formula: radiation intensity of the infarct side (right)/radiation intensity of the unaffected side (left) (r/l, %). For the CbBF analysis, six globular-shaped ROIs (2 × 2 × 2 mm each) were placed in the coronal slices 11.0 mm posterior to the bregma, symmetrically at the medial cerebellar cortex, lateral cerebellar cortex, and cerebellar nuclei. The average of the three ROIs on each side was calculated and the cerebellar perfusion ratio was calculated according to the following formula: average radiation intensity on the side contralateral to the infarct (left)/average radiation intensity on the side ipsilateral to the infarct (right) (l/r, %).

### 4.6. cDNA Microarray

Genes induced by MCAO in the cerebellar cortex were searched comprehensively by microarray. Two days after MCAO, the cerebellar cortical tissues on both sides (contralateral and ipsilateral to the infarct) were collected and homogenized in TRIzol^®^ reagent (Invitrogen, Carlsbad, CA, USA), and total RNA was extracted. In the control group, the left cerebellar cortices were extracted two days after sham surgery. In each group (contralateral, ipsilateral and control group), tissues from four rats were combined to ensure sufficient mass of the experimental material. Sample labeling by Cy3 and array hybridization were performed according to One-Color Microarray-Based Gene Expression Microarrays Analysis (Agilent Technology, Santa Clara, CA, USA). Total RNA from each sample was linearly amplified and labeled with Cy3. Total RNA was checked for quantity using Agilent 2100 Bioanalyze. The Cy3-labeled cRNA was fragmented and hybridized to an Agilent Expression Array (SurePrint G3 Rat Gene Expression 8 × 60 K Ver2.0; Agilent Technology) on which cDNA probes for 30,584 genes had been blotted. This hybridized array was then washed using the Gene Expression Wash Buffer Pack (Agilent Technology) and scanned using the Agilent DNA Microarray Scanner (G2600D). Each gene expression level was calculated from its fluorescence intensity as quantified using Agilent Feature Extraction software. We normalized the gene transcript expression values to the control samples on each chip. The obtained data has been deposited at the NCBI Gene Expression Omnibus (GEO) site (under accession number GSE144547) and is freely available to the scientific community for download and further in-depth analysis. Each gene expression ratio was expressed as the fluorescence intensity of the contra-or ipsilateral group/fluorescence intensity of the control group. Increasing genes were defined as those for which Log2 (ratio) > 1 if the ratio was greater than two, and decreasing genes were defined as those for which Log2 (ratio) < −1 if the ratio was less than one-half. To identify the biological processes or pathways causing global mRNA perturbation, Gene Set Enrichment Analysis (GSEA, v4.0.2, Broad Institute, MA, USA) was performed to assess the enrichment of signature gene sets from the contralateral and control groups. In the present study, GSEA was performed using the collected “Hallmarks” gene-sets with the following parameters: 1000 gene set permutations, gene set size between 15 and 500, weighted enrichment statistics.

### 4.7. Western Immunoblotting Using Brain Homogenates

Two days after the surgery, rats from the MCAO group and the control group (*n* = 6 per group) were processed with western immunoblotting. From the MCAO group rats, each side of the cerebellar cortex (i.e., that contralateral to the infarct (left hemisphere) and that ipsilateral to the infarct (right hemisphere)) was extracted as a separate sample. From the control group rats, the left cerebellar cortex was extracted. Sample preparation and western blotting were performed as described previously [33]. The expression of nuclear factor erythroid 2-related factor 2 (Nrf2) and heme oxygenase-1 (HO-1) in the cerebellar cortex was evaluated. The antibodies included anti-Nrf2 antibody (1:1000, ab137550; abcam), anti-HO-1 antibody (1:1000, ab68477; abcam), and anti-beta-actin antibody (1:5000, A5441; Sigma-Aldrich, St. Louis, MO, USA), with anti-mouse and anti-rabbit IgG HRP-linked secondary antibodies (both 1:5000, Cell Signaling Technology, Danvers, MA, USA). Signals were analyzed using the VersaDoc molecular imaging system (Bio-Rad, Hercules, CA, USA), and protein levels were normalized to beta-actin.

### 4.8. TUNEL Analysis

In order to evaluate apoptosis in the cerebellum, TUNEL assay staining was performed in a new cohort of 11 rats using a commercial kit according to the manufacturer’s instructions (No. 8442, Medical & Biological Laboratories, Nagoya, Japan). Rats were sacrificed seven days after MCAO. The brain tissues were rapidly removed and postfixed in formalin. After postfixed tissues were embedded in paraffin wax, a 6-μm-thick coronal section was obtained from each sample 11.0 mm posterior to the bregma. The deparaffinized sections were incubated with TdT enzyme, which links digoxigenin-deoxyribose nucleoside triphosphate (dNTP) to apoptotic DNA fragments. Anti-digoxigenin antibody conjugated with fluorescein was applied to detect the digoxigenin-dNTP tails. After TUNEL assay, sections were counterstained with propidium iodide (PI) (Life Technologies, Carlsbad, CA, USA). TUNEL+/PI+ cells in the entire region were counted at 20× magnification. The data were expressed as the number of TUNEL+/PI+ cells in each cerebellar hemisphere.

### 4.9. Statistical Analysis

At the start of the study, the sample size estimation for each protocol was calculated using G*Power software version 3.1 (Heinrich Heine University, Düsseldorf, Germany) based on the result of the preliminary experiments. All data in the study are presented as mean ± standard deviation. Statistical significance was assessed by two-tailed t test for comparisons between two groups. Data ware analyzed using JMP software version 10.0.2 (SAS Institute, Cary, NC, USA) and a *p*-value less than 0.05 was considered to indicate significance.

## 5. Conclusions

Gene expression analysis showed that oxidative stress, apoptosis, and hypoxia may be related to the pathophysiology of CCD in a rat MCAO model.

## Figures and Tables

**Figure 1 ijms-21-04137-f001:**
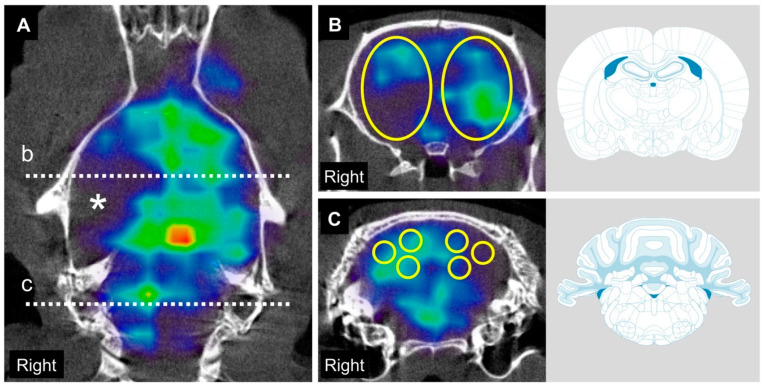
Single photon emission computed tomography (SPECT)/computed tomography (CT) images and regions of interest (ROI) settings to evaluate cerebral and cerebellar blood flow after MCAO. SPECT acquisitions were color-coded with National Institute of Health spectrum color scale. (**A**), Vertical section of SPECT/CT image obtained 2 days after middle cerebral artery occlusion (MCAO) shows obvious hypointensity at the right middle cerebral artery territory (*). Dashed line “b” indicates the coronal slice 2 mm posterior from bregma and dashed line “c” indicates the coronal slice 11 mm posterior from bregma. (**B**) In the coronal slice 2 mm posterior from bregma, 6 × 9 × 6 mm 3D-elliptical ROIs were symmetrically placed. (**C**) In the coronal slice 11 mm posterior from bregma, 2 × 2 × 2 mm global ROIs were symmetrically placed at medial cerebellar cortex, lateral cerebellar cortex and cerebellar nuclei of each side respectively.

**Figure 2 ijms-21-04137-f002:**
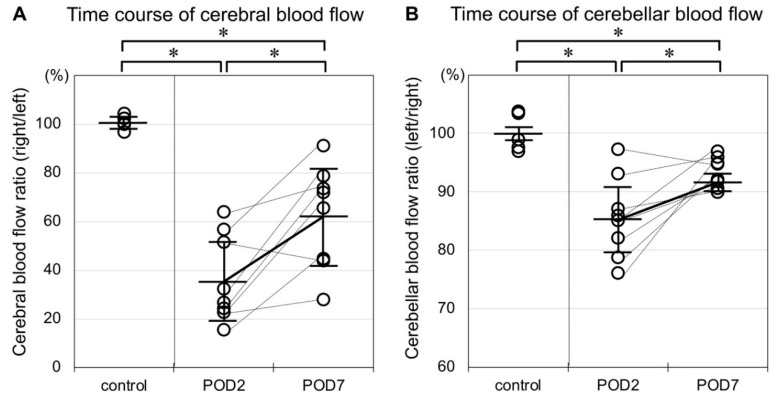
(**A**) Rats in the control group showed no apparent laterality of cerebral blood flow. Cerebral blood flow ratio (right/left) was significantly decreased two and seven days after MCAO. The plots of each individual are connected with bottled lines. This ratio significantly recuperated over time. (**B**) Rats in the control group showed no apparent laterality of cerebellar blood flow. Cerebellar blood flow ratio (left/right) was significantly decreased two and seven days after MCAO, but this decrease was of a lesser degree than that in the cerebrum. As in the cerebrum, the decrease in blood flow ratio in the cerebellum likewise recuperated over time. (* *p* < 0.05, *n* = 5 in the control group, *n* = 8 in the MCAO group).

**Figure 3 ijms-21-04137-f003:**
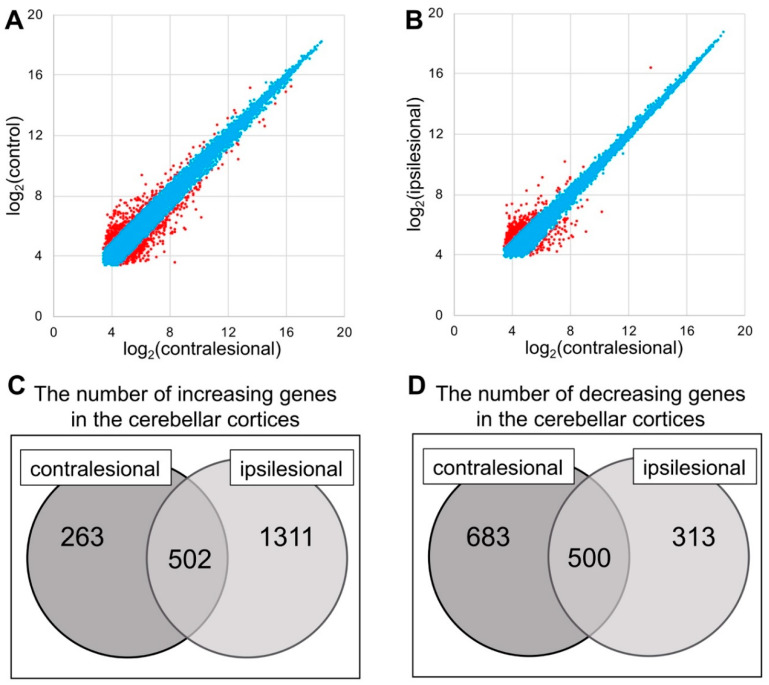
(**A**,**B**) Correlation plot of the contralesional, ipsilesional and control group. The genes with expression change between two conditions <2.0× are labeled as blue and ≥2.0× as red. (**C**,**D**) The mRNA gene expression ratio was expressed as the ratio of the fluorescence intensity of the contra-or ipsilesional group to that of the control group. Increasing genes were defined as those for which Log2 (ratio) > 1 if the ratio was more than double, while decreasing genes were defined as those for which Log2 (ratio) < −1 if the ratio was less than one-half.

**Figure 4 ijms-21-04137-f004:**
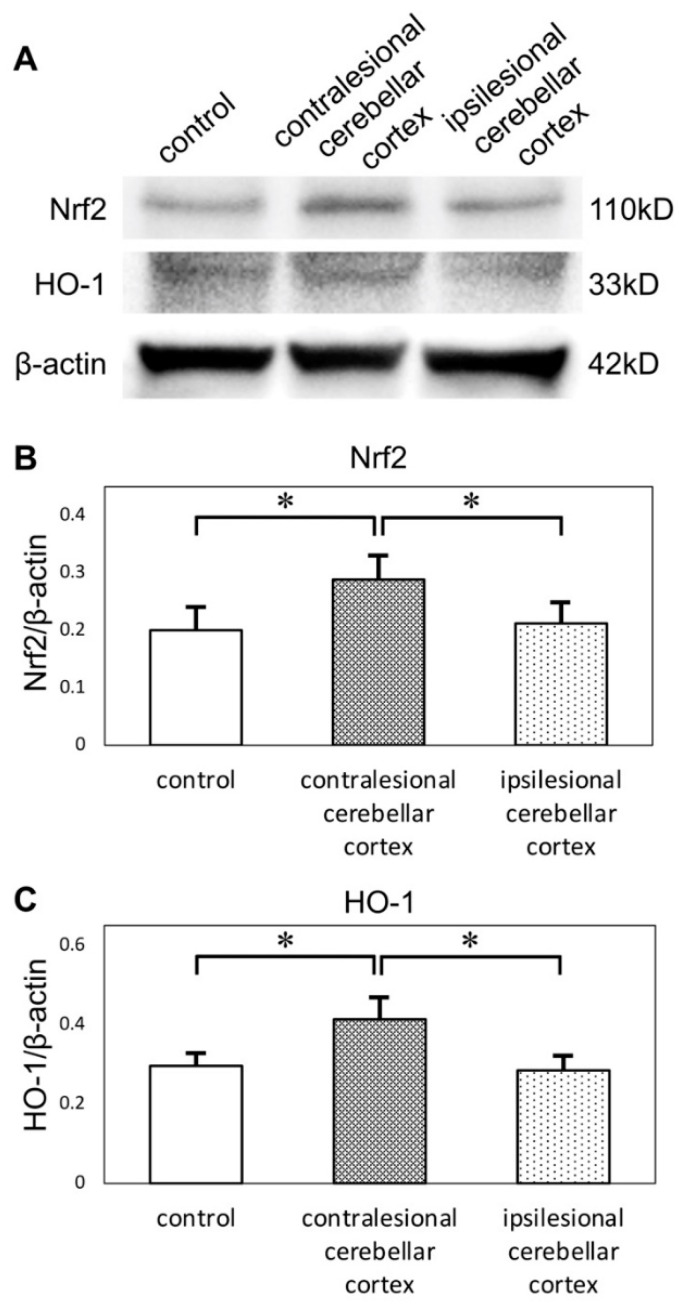
(**A**) Western blots of cerebellar cortices show expression of Nrf2 and HO-1. Protein levels were normalized to beta-actin. (**B**) Quantification of western blots by densitometric analysis indicated that the expression of Nrf2 was upregulated in contralesional cerebellar cortices. (**C**) The expression of HO-1 was also upregulated in contralesional cerebellar cortices (* *p* < 0.05 versus other groups, *n* = 6 in each group). Nrf2: nuclear factor erythroid 2-related factor 2, HO-1: heme oxygenase-1.

**Figure 5 ijms-21-04137-f005:**
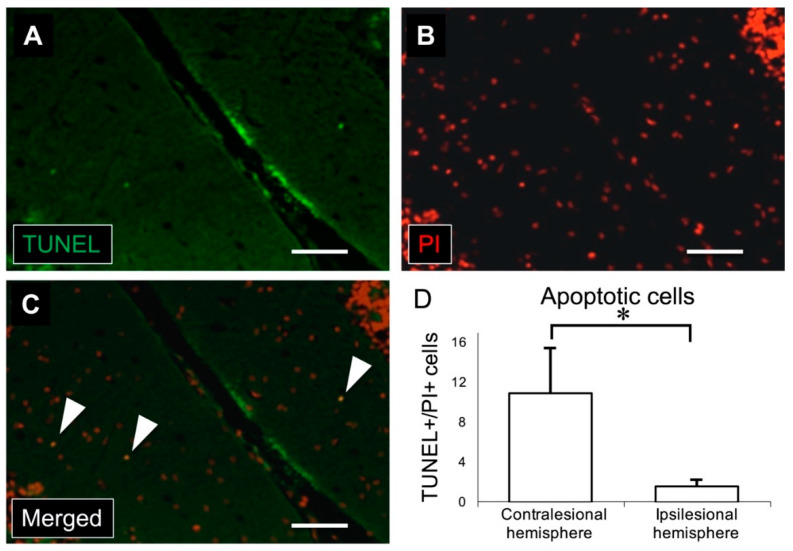
(**A**) Immunostaining for TUNEL (green) shows apoptotic cells in the molecular layer of the cerebellar cortex. (**B**) Nuclei were stained with PI in red. (**C**) Merged image of TUNEL and PI immunostaining. White arrowheads indicate TUNEL/PI-positive cells. (**D**) There was a significant increase in the number of TUNEL/PI-positive cells in the contralesional (left) cerebellar cortex compared to the ipsilesional (right) cerebellar cortex (scale bar: 50 μm, * *p* < 0.001, *n* = 11 in each group). TUNEL: terminal deoxynucleotidyl transferase deoxyuridine triphosphate nick-end labeling, PI: propidium iodide.

**Table 1 ijms-21-04137-t001:** Upregulated gene sets of the contralesional cerebellar cortex.

Gene Sets	NES	Nominal *p*-Value	FDR *q*-Value
Apical surface	1.4829147	0.08070175	0.41270936
Oxidative phosphorylation	1.4047105	0.03902439	0.3811725
Hypoxia	1.3170835	0.056603774	0.421758
Apoptosis	1.2919638	0.05142857	0.36732572
Reactive oxygen species pathway	1.1763533	0.20440252	0.56658465
Fatty acid metabolism	1.1711965	0.1904762	0.4835092
IL6 JAK STAT3 signaling	1.161279	0.19491525	0.43585464
p53 pathway	1.1298119	0.21857923	0.45179152
Interferon-γ response	0.999022	0.4489796	0.7390757
TNF-α signaling via NF-κB	0.93690044	0.61290324	0.85676676
UV response UP	0.81916744	0.8905473	1
Myogenesis	0.79656494	0.9	1
Spermatogenesis	0.7820479	0.85542166	1
Glycolysis	0.76751935	0.90338165	0.95213044
DNA repair	0.65829355	0.97890294	0.97356385

NES indicates normalized enrichment score; FDR, false discovery rate.

**Table 2 ijms-21-04137-t002:** Downregulated gene sets of the contralesional cerebellar cortex.

Gene Sets	NES	Nominal *p*-Value	FDR *q*-Value
Angiogenesis	−1.3574327	0.08066759	1
Apical junction	−1.3069164	0.0870098	1
Bile acid metabolism	−1.2624685	0.1498029	1
G2M checkpoint	−1.2126812	0.19066148	1
KRAS signaling DN	−1.1927822	0.1927555	1
Xenobiotic metabolism	−1.1860862	0.18062201	1
Myc targets v1	−1.1843342	0.23136246	0.89537966
UV response DN	−1.1636689	0.21843435	0.87870497
coagulation	−1.153253	0.26289308	0.8315621
Heme metabolism	−1.1413887	0.27120823	0.79227114
Estrogen response early	−1.1271846	0.25159642	0.7696532
Mitotic spindle	−1.1119729	0.31737345	0.7581073
Androgen response	−1.0270199	0.44093406	0.9964891
KRAS signaling UP	−0.98356724	0.5177665	1
Allograft rejection	−0.95813197	0.57441574	1
E2f targets	−0.94351673	0.5536424	1
Cholesterol homeostasis	−0.9269579	0.58760107	1
Peroxisome	−0.8992629	0.6364847	1
Epithelial mesenchymal transition	−0.88842046	0.6766467	1
IL2 STAT5 signaling	−0.87723947	0.6804878	1
TGF β signaling	−0.8657552	0.67280453	1
Inflammatory response	−0.8562549	0.7315036	0.9768945
Adipogenesis	−0.83410215	0.78725964	0.98129094
Complement	−0.7720234	0.86107785	1
Estrogen response late	−0.7568291	0.88578373	1
Interferon α response	−0.73042226	0.83664775	1
PI3K AKT mTOR signaling	−0.6855016	0.92736703	1
mTORC1 signaling	−0.644657	0.9788294	1
Protein secretion	−0.59601074	0.9776021	1
Unfolded protein response	−0.457154	1	0.99926555

NES indicates normalized enrichment score; FDR, false discovery rate.

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
