# Peer review of "Cerebellar Blood Flow and Gene Expression in Crossed Cerebellar Diaschisis after Transient Middle Cerebral Artery Occlusion in Rats"

_ijms, 2020, doi:10.3390/ijms21114137_

Round 1

Reviewer 1 Report

Title: Cerebellar blood flow and gene expression in crossed cerebellar diaschisis after transient middle cerebral artery occlusion in rats

In this manuscript, the authors examined the cerebellar blood flow and gene expression changes in cerebellum after transient MCAO. Specifically, the authors identified a significant reduction of cerebellar blood flow at 2 and 7 days after injury. Gene enrichment analysis helped to identified list of genes associated with oxidative phosphorylation, hypoxia, apoptosis, and reactive oxygen species pathway regulated after injury. Further, authors validated that the protein expressions of Nrf2 and HO-1 were upregulated. Finally, TUNEL staining revealed increased numbers of apoptotic cell numbers post injury. Overall, this study provided useful information of using MCAO model to study the pathogenesis of crossed cerebellar diaschisis and revealed important pathways enriched. However, certain suggested revisions are required and listed below.

Critiques for revision:

  1. The authors are suggested to justify the rationale of examining Nrf2 and HO-1 after injury. Are those genes also significantly upregulated in the gene sequencing analysis?
  2. Why only rats with mNSS between 7 and 12 at one day post injury were selected?
  3. For the discussion on Nrf2 and HO-1, the authors are suggested to cite other relevant references, including PMID: 27734935 and 31900786.

Author Response

Response to Reviewer 1

Thank you so much for your time and effort to review our manuscript. We revised our manuscript following comments from reviewers. Responses to your suggestion/comment are addressed in the attached document.

Please note that the page and line number in the response to the comments are indicated based on the revised Word file, NOT showing “Track Changes” function.

Reviewer 2 Report

Single-photon emission computed tomography (SPECT) makes it possible to visualize physiological and chemical processes in the cerebrum. N-isopropyl-(123I)-p-iodo-amphetamine (123I-IMP) it is a certified radio-ligand well tested in human studies.

Actually few studies were realized on rats models.

Here, the authors use the SPECT technique to evaluate changes in cerebellar blood flow after MCAO lesion in rats. They perform experiments two-time points: 2 and 7 days after MCAO. Their scope demonstrates that MCAO lesion is reversible in 7 or more days, but the lesion is able to downregulate and upregulate many important genes and nuclear factors.

This paper represents a novelty in the re-construction of the post-ischemic phases, and although this analysis is not supported by functional result (as like as cell metabolism changes, or electrophysiological changes in neuronal population), the paper still very interesting.

I personally very appreciated that the authors have discussed the study limitations, especially with regard to the time window used for the experiments. IT was very important to known the genes setting after two months, when damage was consolidated.

Anyway, they showed important data relative to reperfusion damage. However, it was more important to analyze the gene set of “apoptosis”. I believe that the paper would make a qualitative leap if the authors showing the involvement of caspases -3, -6, and -7 (effector caspases), by immuno-histochemical or biochemical analysis (suggestion: Asian J Neurosurg. 2011 Jul-Dec; 6(2): 72–77.). There are many pieces of evidence that link the expression of NR2B and caspase-3 expression.

The authors write that 44 Adult male Wistar rats were used. How many animals died after MCAO?

Did the authors perform a power analysis before starting the project? How many rats were hypothesized for each condition?

 Minor point:

  • In figure 5 panel D, to help the readers should be replaced left and right with contralesional and ipsilesional hemisphere and use left and right only in the legend.
  • In the figure legend sometimes is highlighted the exact p-value, while sometimes is showed the only p> or p<; please standardize it in all legend.

Author Response

Response to Reviewer 2

Thank you so much for your time and effort to review our manuscript. We revised our manuscript following comments from reviewers. Responses to your suggestion/comment are addressed in the attached document.

Please note that the page and line number in the response to the comments are indicated based on the revised Word file, NOT showing “Track Changes” function.

Reviewer 3 Report

In this paper the authors evaluated the chronological changes in CCD (crossed cerebellar diaschisis) using single photon emission computed tomography (SPECT) in a rat middle cerebral artery occlusion model, and they examined gene expression patterns in the cerebellar hemispheres in CCD, and performed histopathological analysis.

This is a very interesting and good graphically developed study

In my opinion some questions should have been asked.

Major/important questions

  1. Introduction:

The introduction is too short. It has to be expanded.

  1. Material and method

The division of animals into experimental groups is poorly described, I couldn't find what is the control group for researchers (how many mice were used and what were done with them before sacrificing)?

Author Response

Response to Reviewer 3

Thank you so much for your time and effort to review our manuscript. We revised our manuscript following comments from reviewers. Responses to your suggestion/comment are addressed in the attached document.

Please note that the page and line number in the response to the comments are indicated based on the revised Word file, NOT showing “Track Changes” function.

Round 2

Reviewer 2 Report

The manuscript is improved as suggested. The study of caspase 3 would take the manuscript to a higher level. Anyway, I can understand that this part of the study could be long and difficult given the number of animals required.

So I think the manuscript can be published as a preliminary study but it is necessary to insert the marking "preliminary data" IN DISCUSSION.

Author Response

Response to Reviewer 2

Thank you again for your comments. We revised our manuscript following your comments and responses are addressed below.

> So I think the manuscript can be published as a preliminary study but it is necessary to insert the marking "preliminary data" IN DISCUSSION.

Thank you for your suggestion. We added following sentence in the “Discussion” section.

Page 9 Line 201-203
Since this study could be a preliminary data, further investigation is needed to clarify the molecular mechanisms of deafferentation and their relationship to oxidative stress and apoptosis.